# Microbial Landscape and Antibiotic-Susceptibility Profiles of Microorganisms in Patients with Bacterial Pneumonia: A Comparative Cross-Sectional Study of COVID-19 and Non-COVID-19 Cases in Aktobe, Kazakhstan

**DOI:** 10.3390/antibiotics12081297

**Published:** 2023-08-08

**Authors:** Nurgul Ablakimova, Aigul Z. Mussina, Gaziza A. Smagulova, Svetlana Rachina, Meirambek S. Kurmangazin, Aigerim Balapasheva, Dinara Karimoldayeva, Afshin Zare, Mahdi Mahdipour, Farhad Rahmanifar

**Affiliations:** 1Department of Pharmacology, West Kazakhstan Marat Ospanov Medical University, Aktobe 030012, Kazakhstan; a.mussina@zkmu.kz (A.Z.M.); g.smagulova@zkmu.kz (G.A.S.); a.balapasheva@zkmu.kz (A.B.); 2Hospital Therapy Department No. 2, I.M. Sechenov First Moscow State Medical University, Moscow 119435, Russia; rachina_s_a@staff.sechenov.ru; 3Department of Infectious Disease, West Kazakhstan Marat Ospanov Medical University, Aktobe 030012, Kazakhstan; m.kurmangazin@zkmu.kz; 4Respiratory Medicine and Allergology Department, Aktobe Medical Center, Aktobe 030017, Kazakhstan; karimoldaeva@mail.ru; 5PerciaVista R & D Co., Shiraz 71676-83745, Iran; afshinzareresearch@gmail.com; 6Stem Cell Research Center, Tabriz University of Medical Sciences, Tabriz 51666-53431, Iran; mahdi.mahdipour@gmail.com; 7Department of Applied Cell Sciences, Faculty of Advanced Medical Sciences, Tabriz University of Medical Sciences, Tabriz 51666-53431, Iran; 8Department of Basic Sciences, School of Veterinary Medicine, Shiraz University, Shiraz 71348-14336, Iran; f.rahmanifar@yahoo.com

**Keywords:** pneumonia, bacterial, etiology, sputum, pathogens, COVID-19, antibiotics, antimicrobial resistance

## Abstract

This cross-sectional study investigated the microbial landscape and antibiotic-resistance patterns in patients with bacterial pneumonia, with a focus on the impact of COVID-19. Sputum samples from individuals with bacterial pneumonia, including coronavirus disease 2019-positive polymerase chain reaction (COVID-19-PCR+), COVID-19-PCR− and non-COVID-19 patients, were analyzed. Surprisingly, the classic etiological factor of bacterial pneumonia, *Streptococcus pneumoniae*, was rarely isolated from the sputum samples. Furthermore, the frequency of multidrug-resistant pathogens was found to be higher in non-COVID-19 patients, highlighting the potential impact of the pandemic on antimicrobial resistance. Strains obtained from COVID-19-PCR+ patients exhibited significant resistance to commonly used antibiotics, including fluoroquinolones and cephalosporins. Notably, the ESKAPE pathogens, *Staphylococcus aureus*, *Klebsiella pneumoniae*, *Pseudomonas aeruginosa*, *Enterobacter cloacae*, and *Enterobacter aerogenes*, were identified among the isolated microorganisms. Our findings underscore the urgent need for infection control measures and responsible antibiotic use in healthcare settings, as well as the importance of enhancing pneumonia diagnostics and implementing standardized laboratory protocols.

## 1. Introduction

Pneumonia, a respiratory infection associated with significant morbidity and mortality [1], poses a global health challenge, despite progress in the field of antimicrobial treatments, diagnostic techniques for microbiological identification, and preventive strategies. It is one of the leading causes of sepsis [2] and is responsible for a significant number of calls to emergency departments worldwide, especially during the coronavirus disease 2019 (COVID-19) pandemic [3]. According to Dagenais et al. [4], respiratory tract diseases account for 8.5% of all hospital admissions, with 0.5% resulting in mortality. A study of the global burden of diseases, injuries, and risk factors conducted in 2017 revealed that lower respiratory tract infections, including bacterial pneumonia, result in nearly 2.56 million deaths across various age groups [5]. Additionally, hospital-acquired pneumonia (HAP) is one of the most common healthcare-associated infections contributing to death and is estimated to increase hospital stays by up to 12 days and durations of mechanical ventilation by up to 10 days [6].

Different pathogens have been identified as causes of pneumonia, including bacteria, viruses, and fungi. However, only a small number of these pathogens are responsible for the majority of cases in individuals with a healthy immune system [7]. Among the viral pathogens, influenza and rhinovirus have been identified as causes in approximately one-third of community-acquired pneumonia (CAP) cases [8]. However, confirming the viral etiology has been challenging in the past due to the frequent co-occurrence of bacterial and viral infections [9] and the possibility that viruses may sometimes be present as colonizers rather than true pathogens. Numerous studies have reported that typical pneumonia is primarily caused by *Streptococcus pneumoniae*, *Staphylococcus aureus*, *Klebsiella pneumoniae*, *Haemophilus influenzae*, *Pseudomonas aeruginosa*, *Moraxella catarrhalis*, and *Escherichia coli* [2] while atypical pneumonia is predominantly caused by *Legionella pneumophila*, *Chlamydia pneumoniae*, and *Mycoplasma pneumoniae* [10]. Although *S. pneumoniae* is the most prevalent pathogen causing CAP worldwide across all age groups, gram-negative bacteria, such as *K. pneumoniae*, *Acinetobacter baumannii*, *P. aeruginosa*, and *E. coli*, are commonly associated with HAP [11].

The rapidly escalated COVID-19 pandemic has focused attention on the diagnosis and treatment of patients with acute respiratory infections. Furthermore, it is reported that many ‘suspected’ cases displaying typical clinical characteristics of COVID-19 and identical, specific computed tomography images were not diagnosed by using the polymerase chain reaction (PCR) method, considered the ‘gold standard’ [12]. Moreover, identifying coinfecting pathogens is crucial in the treatment of COVID-19 patients as coinfection with bacterial pathogens has been associated with increased disease severity and mortality [13,14]. Therefore, the accurate detection and management of bacterial coinfections are of great significance in the overall treatment approach for COVID-19-infected individuals. The efficacy of antibiotics used in the treatment of bacterial pneumonia has been compromised due to the increasing prevalence of resistant pathogens [15]. As the microbial landscape and antibiotic susceptibility of pathogens involved in bacterial pneumonia continue to evolve, it is crucial to investigate and understand these changes.

The most reliable method for identifying the cause of pneumonia is detecting respiratory pathogens in samples collected directly from the affected area, specifically the lungs [16]. This can be accomplished through procedures such as bronchoalveolar lavage, obtaining samples of pleural fluid, or performing a lung biopsy [17]. Despite this, sputum testing remains the most commonly used method for detecting the etiology of pneumonia in Kazakhstan, given its origin from the LRT and the non-invasive nature of the sample collection. Therefore, this study aims to compare the characteristics of pathogens isolated from the sputum samples of patients with COVID-19 with positive for PCR (PCR+) or negative for PCR (PCR−) and those with bacterial pneumonia without COVID-19 (non-COVID-19). Additionally, we assessed the level of pathogen resistance to antimicrobial drugs.

## 2. Results

### 2.1. Population Characteristics and Frequency of Multidrug Resistance

This research investigated the findings of a sputum analysis performed on individuals receiving treatment for bacterial pneumonia at hospitals in the city of Aktobe between 2021 and 2022. The analysis involved patients who had bacterial pneumonia as their primary ailment, as well as those who had bacterial pneumonia as a comorbidity. The study population was categorized into three groups: COVID-19 patients who had typical clinical characteristics of COVID-19 and radiological data and tested PCR+, COVID-19 patients who also had typical clinical characteristics and radiological data but tested PCR−, and non-COVID-19 patients with bacterial pneumonia without signs of COVID-19 infection and negative results for the PCR. The analysis revealed that non-COVID-19 patients with bacterial pneumonia had a higher frequency of isolating multidrug-resistant (MDR) microorganisms from their sputum samples compared to patients with COVID-19 (Table 1).

### 2.2. Most of the Microorganisms Which Were Identified Were Not the Main Etiological Factors of Pneumonia

Out of the 340 samples analyzed, all showed positive cultures. The samples were obtained from adult patients. Mixed cultures were detected in 6.18% (21) of the samples, all of which were collected from non-COVID-19 patients (25% of the total). A total of 35 distinct microorganisms were identified, with gram-negative bacteria accounting for 46.8%, gram-positive bacteria for 37.6%, and fungi for 15.6% of the cases. The distribution of each identified pathogen is illustrated in Figure 1.

### 2.3. Enterobacterales and S. aureus Were Found Most Often in COVID-19 and Non-COVID-19 Patients

In 59% of cases in patients with the COVID-19 infection, potential pneumonia pathogens were isolated; meanwhile, in 41% of cases, non-pathogenic microorganisms or commensals were identified, mainly *Candida* spp. and *S. epidermidis*. A similar situation was observed in non-COVID-19 patients, where the main causes of pneumonia were found in 55% of all cases. The comparison of the potential pathogens’ frequencies is shown in Figure 2.

During the study and analysis of the results, it was observed that the classic etiological factor of bacterial pneumonia, *S. pneumoniae*, was rarely isolated from the sputum samples, only occurring in one case. *K. pneumoniae*, the predominant microorganism, was isolated in 17.65% of all cases. There was no significant difference in the frequency of this pathogen among the three groups. Among COVID-19-PCR− patients, the most commonly isolated microorganism was *S. aureus*, accounting for 20.3% of cases. In non-COVID-19 patients, the frequency of *S. aureus* was lower compared to the COVID-19 groups.

### 2.4. The ESKAPE Pathogens S. aureus, K. pneumoniae, P. aeruginosa, E. cloacae, and E. aerogenes were Identified among the Pathogens Analyzed

In February 2017, the World Health Organization (WHO) published a list of pathogens that necessitate the urgent development of novel antimicrobial drugs, providing guidance and prioritizing research and development endeavors. Within this comprehensive list, a group of pathogens known as ESKAPE pathogens (comprising *Enterococcus faecium*, *S. aureus*, *K. pneumoniae*, *A. baumannii*, *P. aeruginosa*, and *Enterobacter* spp.) were specifically recognized as having “priority status” [18]. Our study identified *S. aureus*, *K. pneumoniae*, *P. aeruginosa*, *E. cloacae*, and *E. aerogenes* among the pathogens analyzed. However, *A. baumannii* and *E. faecium* were not detected. The resistance profiles of gram-negative pathogens, belonging to the ESKAPE group, isolated from the sputum of COVID-19 and non-COVID-19 patients are presented in Figure 3.

### 2.5. Gram-Negative Bacteria Obtained in COVID-19 and Non-COVID-19 Patients Showed Different Degrees of Resistance to Commonly Used Antibiotics

Among the microorganisms isolated from the sputum samples of patients with bacterial pneumonia, 25.6% were identified as members of the *Klebsiella* genus. These pathogens showed a high level of resistance to levofloxacin (51.5% in COVID-19 patients, 29.4% in non-COVID-19 patients), as well as to ciprofloxacin (32.9% in COVID-19 patients, 27.3% in non-COVID-19 patients) and amikacin (27.1% in COVID-19 patients, 47% in non-COVID-19 patients). COVID-19 patients also exhibited significant resistance to cefepime (51.4%). *Citrobacter diversus*, which was detected only in the sputum samples of patients with COVID-19 (*n* = 17), showed high resistance to fluoroquinolones and cephalosporins: levofloxacin 68.8%, ciprofloxacin 58.8%, cefepime 52.9%.

## 3. Discussion

Most of the pathogens which were identified were not the main etiological factors of pneumonia. During the study and analysis of the results, it was observed that the classic etiological factor of bacterial pneumonia, *S. pneumoniae*, was rarely isolated from the sputum samples. The wide range of these figures can be attributed to difficulties in obtaining high-quality sputum samples from the lower respiratory tract, differences in the sensitivity of diagnostic media and tests used, and the prior use of antimicrobial agents for treating the disease before an etiological diagnosis is conducted. These factors contribute to the variability in the detection rates of *S. pneumoniae* [19]. However, similar findings were reported by Lavrinenko et al. [20], where, in only 1% of the cases (specifically three out of two-hundred-nine patients from Almaty, Karaganda and Atyrau, Kazakhstan) with coronavirus infection included in the study were they found to have a pneumococcal infection. Physical distancing and other measures used during COVID-19 can be some of the reasons for the significant reduction in invasive diseases caused by *S. pneumoniae* [21]. These results show that the quality and technique used for collecting sputum samples can greatly influence the detection of bacterial pathogens.

The frequency of multidrug-resistant pathogens’ isolation was higher after the COVID-19 pandemic. Our research supports this observation as we noticed an increased incidence of bloodstream infections and higher mortality rates following MDR infections during the post-COVID-19 period. In fact, 68.8% of bloodstream cultures showed MDR bacteria, compared to 40.0% in the pre-COVID-19 period [22]. Supporting our findings, La et al. [23] also reported a significant rise in the prevalence of MDR organisms after the COVID-19 pandemic, particularly in cases of bacteremia in South Korean hospitals. This surge may be attributed to the inappropriate use of antibiotics during the pandemic. Despite a low overall proportion of bacterial coinfections among COVID-19 patients, the usage of antibiotics was high [24]. These findings indicate that the improper use of antibiotics during the pandemic, despite the low overall proportion of bacterial coinfections among COVID-19 patients, might have contributed to the emergence and spread of multidrug-resistant strains.

*S. aureus*, *S. epidermidis*, *K. pneumoniae*, and *C. albicans* were found most often in patients with COVID-19. In line with previous studies, *S. aureus* was categorized as an emerging copathogen in individuals affected by COVID-19 [25]. The isolation of *Candida* fungi from the sputum of COVID-19 patients also has been reported by other authors [20,26,27,28,29,30]. There is a significant need for a cautious interpretation of fungal isolates from respiratory samples, especially in non-critically ill patients with a low pre-test probability of invasive fungal infections due to poor oral hygiene [31,32,33], immune dysregulation [34], and viral cytopathic effects on epithelial cells [35]. The frequent isolation of commensals may also indicate shortcomings in laboratory diagnostics, particularly the need for careful evaluation of the quality of the provided biomaterial and the mandatory performances of microscopic examination and gram staining. Further research and careful consideration of all of these factors are needed to better understand the clinical implications of these pathogens in the context of COVID-19.

Bacterial strains, including, in particular, the obtained ESKAPE pathogens *S. aureus*, *K. pneumoniae*, *P. aeruginosa,* and *Enterobacter* spp., in patients with COVID-19 showed a high degree of resistance to commonly used antibiotics. The studies conducted by other researchers also emphasize the general pattern of rising antimicrobial resistance. Among the 159 strains of bacteria isolated from COVID-19 patients in Wuhan, China, 85.5% were gram-negative, with *A. baumannii* (35.8%), *K. pneumoniae* (30.8%), and *Stenotrophomonas maltophilia* (6.3%) as the top three bacteria causing bloodstream infections while carbapenem resistance was high in *A. baumannii* (91.2%) and *K. pneumoniae* (75.5%) and meticillin resistance was observed in all *S. aureus* and coagulase-negative staphylococci [36]. In another study conducted by Fu et al., it was found that five critically ill COVID-19 patients experienced secondary infections caused by extended-spectrum beta-lactamase-producing *K. pneumoniae*, *S. maltophilia*, *Burkholderia cepacia*, and *P. aeruginosa*, which were identified as the responsible pathogens [37]. Table 2 compares the effect of COVID-19 on the landscape and the level of resistance change during the pandemic in our study and previous ones. The analysis of articles from Web of Knowledge using the search terms “covid”, “antibiotic resistance”, and “respiratory” in the abstract or title led us to find 55 relevant articles, of which a 10th of them evaluated the effect of COVID-19 on landscape change in antibiotic resistance. Between them, eight articles demonstrated the impact of COVID-19 on these changes.

In contrast to our findings, Zeshan et al. [48] reported that isolated strains of *K. pneumoniae* in 88%, *P. aeruginosa* in 75%, and *S. aureus* in 45% of cases were resistant to ciprofloxacin. Other researchers also noted that there is high resistance to fluoroquinolone and cephalosporins among the gram-negative bacteria isolated from the COVID-19 patients [41,49]. This may be a reflection of the widespread use of these medications for empirical therapy. As indicated by the meta-analysis on bacterial coinfection in COVID-19 [24], a significant proportion of patients (approximately 74%) have received broad-spectrum antibiotics in the form of cephalosporins and fluoroquinolones as an empirical treatment approach. The findings of this study and other research indicate a concerning trend of high antimicrobial resistance among bacterial strains obtained from patients with COVID-19.

There are several limitations of our study that need to be pointed out, starting with those related to the retrospective study design. Firstly, we were limited in our ability to investigate the accuracy of bacterial pneumonia diagnoses and the relationship between the initiation of empirical therapy and the timing of specimen collection, which may have influenced the results. Secondly, it also restricted our ability to examine the role of glucocorticoid medications in the development of resistance. Thirdly, the COVID-19 patient beds in the infectious hospital were temporarily opened and the sputum analysis of non-COVID-19 patients was already being conducted in a different laboratory. However, both laboratories operate according to the same directives and recommendations. The PCR testing of sputum samples was not conducted, thus leaving the role of atypical pathogens unclear in the microbiological landscape. Further studies are required to examine the resistance genes responsible for the observed sensitivity results.

## 4. Materials and Methods

### 4.1. Cases and Ethics

This study was conducted as part of a research project at the West Kazakhstan Marat Ospanov Medical University, entitled “Concomitant bacterial infections and pharmacoepidemiology of antibiotic resistance in patients with COVID-19: the situation in the Aktobe region”. Ethical approval was obtained from the local bioethics committee (Approval No. 8, dated 15 October 2021). A retrospective analysis was carried out, examining 340 medical records of adult patients who were admitted to the COVID-19 wards at the Aktobe Regional Clinical Infectious Hospital between 1 January 2021 and 31 December 2021 and patients who were diagnosed with bacterial pneumonia and treated at the Multidisciplinary Regional Hospital, Aktobe in 2022. The diagnoses of the COVID-19 infection were established based on clinical and epidemiological data, radiological examinations, and the presence of PCR analyses for detecting SARS-CoV-2 RNA from nasopharyngeal swabs. All patients included in this study had documented diagnoses of bacterial pneumonia, according to medical records.

### 4.2. Pathogen Identification

All patients underwent laboratory testing (“Bacteriological culture of sputum with determination of susceptibility to antibacterial agents”) according to standard methods and procedures. After the isolation of a pure culture, the identification of microorganisms was carried out using the MicroScan AutoScan 4 system (SIEMENS, West Sacramento, CA, USA) and MicroScan Rapid system (Beckman Coulter, Sacramento, CA, USA).

### 4.3. Antibiotic Susceptibility

The determination of antimicrobial sensitivity was carried out for the microorganisms in each group of patients. An assessment of the susceptibility of bacterial isolates to antibacterial drugs was carried out by the disk diffusion method. The interpretation of the results was based on the comparison of the growth-inhibition diameter of the tested strains with the tabulated data in accordance with the Clinical and Laboratory Standards Institute (CLSI) recommendations. Internal quality control was performed using reference strains in the microbiology laboratories. A MDR analysis, as defined by the resistance of bacterial strains to three or more groups of antibiotics [50], was conducted to assess the prevalence of MDR among the different patient groups.

### 4.4. Sample Size Calculation

To calculate the sample size to investigate the prevalence of antibiotic resistance in patients with bacterial pneumonia, we selected *K. pneumoniae* in a specific hospital setting. The researchers wanted to estimate the proportion of patients with antibiotic-resistant *K. pneumoniae* with a desired level of precision and statistical power.

Desired Level of Precision: the researchers wanted to estimate the prevalence of antibiotic resistance in *K. pneumoniae* with a 95% confidence level and a margin of error (precision) of ±5%.

Statistical Power: the researchers wanted to achieve a statistical power of 80%, which meant this study would be able to detect a significant difference if it existed.

Methodology for Sample Size Calculation:

Estimating Prevalence: as no previous estimate of the prevalence of antibiotic resistance in this specific setting was available, the researchers decided to use a conservative estimate of 30% based on the available preliminary data analysis, assuming the proportion is evenly split between resistant and non-resistant cases.

Z-score for Confidence Level: for a 95% confidence level, the Z-score corresponding to a 95% confidence level was 1.96.

Margin of Error (Precision): the desired margin of error was ±5%, which translates to 0.05.

Estimating Standard Deviation: as the actual proportion of antibiotic resistance was unknown, the researchers used a conservative estimate of 0.5 for the standard deviation, assuming the highest variability.

Calculating Sample Size: using the formula for sample size calculation for estimating a proportion in a population:n = (Z^2^ × p × (1 − p))/(E^2^)
where:

n = required sample size;

Z = Z-score corresponding to the desired confidence level;

p = estimated proportion of antibiotic resistance;

E = margin of error (precision).

Plugging in the values:

n = (1.96^2^ × 0.3 × 0.5)/(0.05^2^) ≈ 231.

### 4.5. Statistical Analysis

Descriptive statistics were employed to analyze the demographic of the patients’ factors. Continuous variables were presented using the mean ± standard deviation (SD) while categorical variables were represented using frequencies. The distribution of categorical variables was assessed using the X^2^ test. Statistical significance was determined as a p-value below 0.05. Data analysis was carried out using IBM SPSS Statistics 22 (SPSS Inc., Chicago, IL, USA) and GraphPad software (Version 9.5.1, San Diego, CA, USA) was employed for graphical visualization.

## 5. Conclusions

The global outbreak of the coronavirus infection serves as a reminder of the significance of infection control measures and the responsible utilization of antimicrobial agents in healthcare settings. Advancing pneumonia diagnostics requires a multifaceted approach that includes expanding laboratory networks, implementing standardized protocols, enhancing microbiologists’ expertise, and promoting collaboration among stakeholders. By prioritizing these tasks, healthcare systems can strengthen their ability to diagnose pneumonia accurately, manage antimicrobial resistance effectively, and promote rational antibiotic use across all levels of healthcare.

## Figures and Tables

**Figure 1 antibiotics-12-01297-f001:**
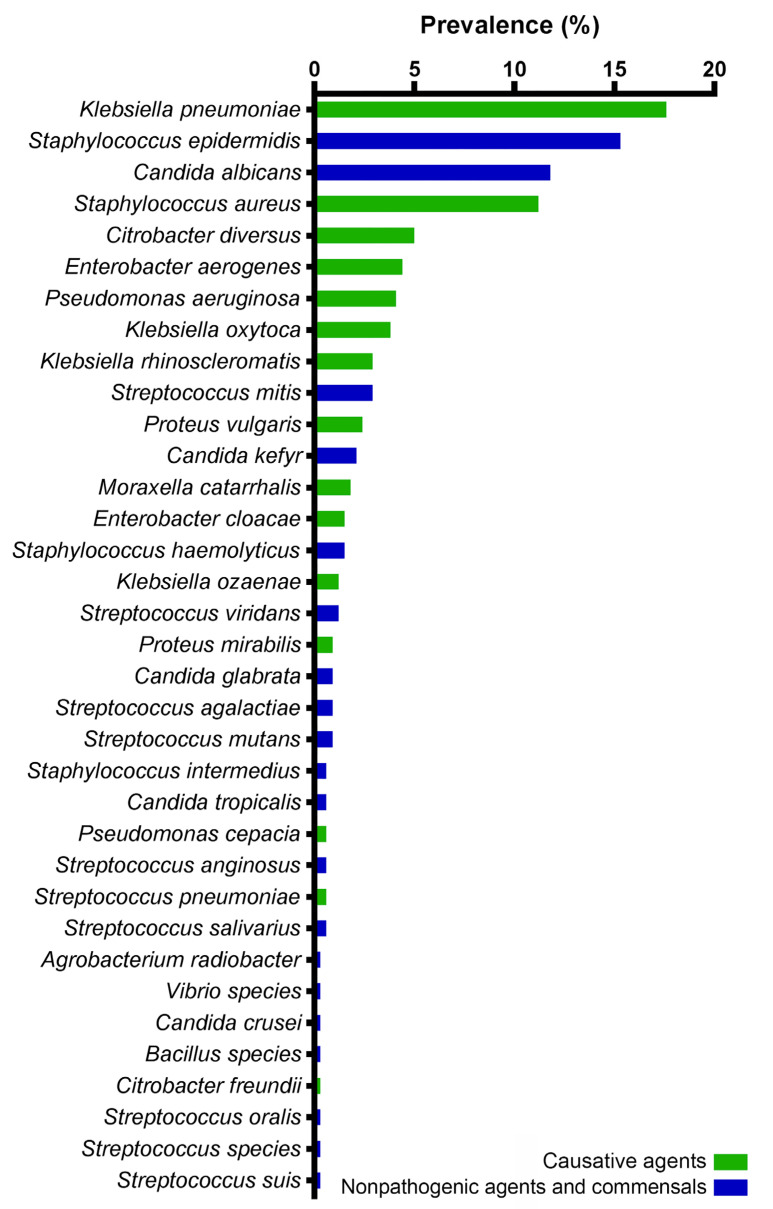
The microbial landscape of the isolated microorganisms from the sputum samples of the patients with COVID-19 and non-COVID-19 bacterial pneumonia in two hospitals in Aktobe, Kazakhstan, 2021–2022.

**Figure 2 antibiotics-12-01297-f002:**
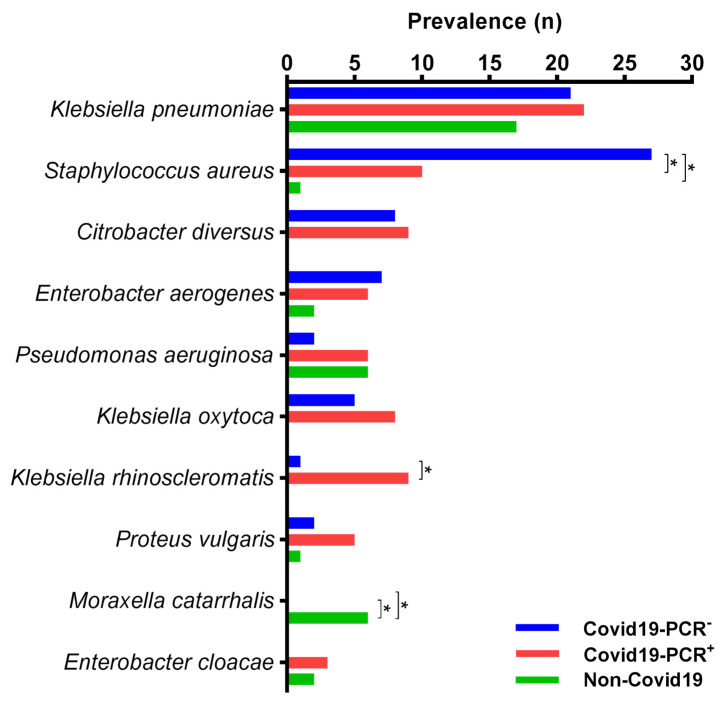
Percentage frequency of bacterial pathogen occurrence from sputum samples of COVID-19-PCR+, COVID-19-PCR−, and non-COVID-19 patients with bacterial pneumonia in two hospitals in Aktobe, Kazakhstan, 2021–2022. * stars show significant differences between columns (*p* < 0.05).

**Figure 3 antibiotics-12-01297-f003:**
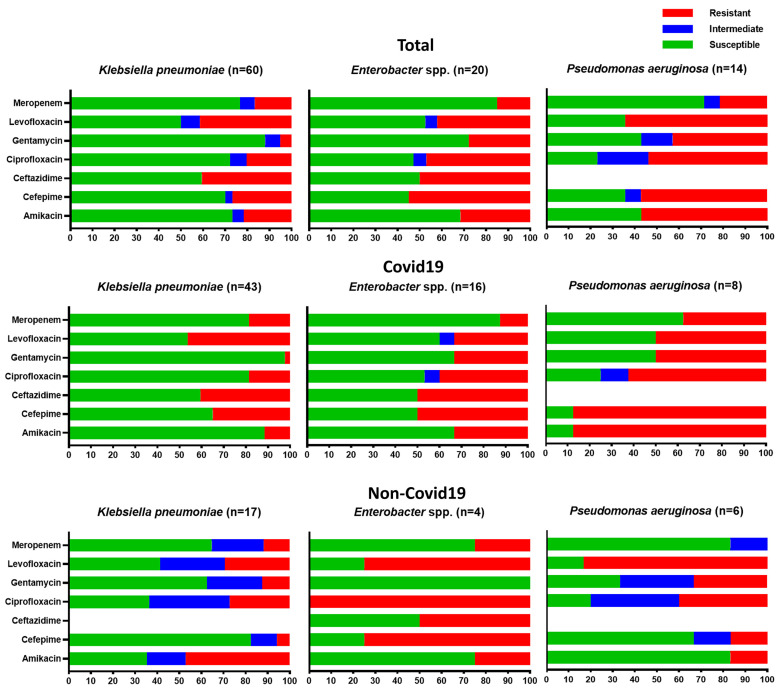
Susceptibility profiles of gram-negative isolates belonging to the ESKAPE pathogen group from sputum samples of patients in two hospitals from COVID-19 and non-COVID-19 patients with bacterial pneumonia in Aktobe, Kazakhstan, 2021–2022.

**Table 1 antibiotics-12-01297-t001:** Characteristics of the studied COVID-19-PCR+, COVID-19-PCR−, and non-COVID-19 patients with bacterial pneumonia in two hospitals in Aktobe, Kazakhstan, 2021–2022.

Factors	COVID-19-PCR+	COVID-19-PCR−	Non-COVID-19
Population, n	133	133	74
Age, year; mean ± SE	59.8 ± 1.3	55.9 ± 1.5	55.2 ± 1.8
Gender			
Male, n (%)	58 (17.1)	62 (18.3)	42 (12.4)
Female, n (%)	75 (22.1)	71 (20.9)	31 (9.1)
Intensive care unit (ICU) treatment, n (%)	31 (9.1)	13 (3.8)	31 (9.1)
Multidrug resistance (MDR), n (%)	66 (19.4)	31 (9.1)	45 (13.2)

**Table 2 antibiotics-12-01297-t002:** Comparisons of the effect of COVID-19 on the landscape and level of antibiotic resistance changes during the pandemic.

Country	No. of Strains	No. of Patients	Landscape and Level of Resistance Change	References
China	18965	ND ^1^	Yes	[38]
India	28	200	No	[39]
India	ND	2000	Yes	[40]
Indonesia	733	2786	Yes	[41]
Iran	192	192	Yes	[42]
Italy	245	157	Yes	[43]
Italy	2002	1090	Yes	[44]
Kazakhstan	340	340	Yes	Current study
Korea	696	1023	No	[45]
Serbia	1410	834	Yes	[46]
USA	31	13	Yes	[47]

^1^ ND, no data.

## Data Availability

Data are available on request due to ethical restrictions.

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
