# Peer review of "Microbial Landscape and Antibiotic-Susceptibility Profiles of Microorganisms in Patients with Bacterial Pneumonia: A Comparative Cross-Sectional Study of COVID-19 and Non-COVID-19 Cases in Aktobe, Kazakhstan"

_antibiotics, 2023, doi:10.3390/antibiotics12081297_

Round 1

Reviewer 1 Report

The article is good but need some improvments. These findings emphasize the urgent need for infection control measures and responsible antibiotic use n healthcare settings. It also highlights the importance of enhancing pneumonia diagnostics and implementing standardized laboratory protocols to effectively identify and treat bacterial pneumonia, especially in the context of the Covid-19

I recommend that a native speaker of English review the manuscript to improve word choice, sentence structure, and grammar.

 As a Reviewer, I would have the following remarks regarding the MS:

        Line 179-190  it is not clear, I recommend to rephrase the entire paragraph.

        Figure 3 it's not clear to read.

        Please describe the abbreviations when you use them for the first time.

In conclusion, advancing pneumonia diagnostics requires a multifaceted approach that includes expanding laboratory networks, implementing standardized protocols, enhancing microbiologists' expertise, and promoting collaboration among stakeholders. By prioritizing these tasks, healthcare systems can strengthen their ability to diagnose pneumonia accurately, manage antimicrobial resistance effectively, and promote rational antibiotic use across all levels of healthcare. Make 3 short conclusion easy to be read!

I recommend that a native speaker of English review the manuscript to improve word choice, sentence structure, and grammar.

Author Response

Responses to the respected reviewer 1:

The article is good but need some improvements. These findings emphasize the urgent need for infection control measures and responsible antibiotic use n healthcare settings. It also highlights the importance of enhancing pneumonia diagnostics and implementing standardized laboratory protocols to effectively identify and treat bacterial pneumonia, especially in the context of the Covid-19

I recommend that a native speaker of English review the manuscript to improve word choice, sentence structure, and grammar.

Response: Thank you for your comment. The article is revised by a native English speaker.

 As a Reviewer, I would have the following remarks regarding the MS:

        Line 179-190  it is not clear, I recommend to rephrase the entire paragraph.

Response: Thank you for your comment. This section is completely paraphrased.

        Figure 3 it's not clear to read.

Response: Thank you for your comment. We revised this figure based on your valuable comment.

        Please describe the abbreviations when you use them for the first time.

Response: Thank you for your suggestion. We described the abbreviations when we use them for the first time.

In conclusion, advancing pneumonia diagnostics requires a multifaceted approach that includes expanding laboratory networks, implementing standardized protocols, enhancing microbiologists' expertise, and promoting collaboration among stakeholders. By prioritizing these tasks, healthcare systems can strengthen their ability to diagnose pneumonia accurately, manage antimicrobial resistance effectively, and promote rational antibiotic use across all levels of healthcare. Make 3 short conclusion easy to be read!

Response: Thank you for your instructive recommendation. We revised the conclusion section based on your valuable suggestion.

 Comments on the Quality of English Language

I recommend that a native speaker of English review the manuscript to improve word choice, sentence structure, and grammar.  

Response: Thank you for your comment. The article is revised by a native English speaker.

Reviewer 2 Report

Overall, I find the manuscript comprehensive and its theme highly interesting and relevant. The quality of the English language used in the article is commendable. I would like to highlight some specific points and offer suggestions for improvement.

Firstly, I appreciate that all the keywords were selected from the MeSH (Medical Subject Headings) terminology, ensuring accuracy and alignment with established medical terminology standards.

Regarding the figures presented in the manuscript, I find them to be of satisfactory quality and meeting the required standards. However, I would suggest revising Figure 3 in terms of drug selection to enhance clarity and accuracy.

In Section 2.3., I recommend placing greater emphasis on pathogens rather than commensals to better align with the objectives of the study.

In Section 2.4., it would be beneficial to include information about the resistance of specific microorganisms and indicate the sensitivity of each one separately. The same correction should be applied to the discussion part, specifically on line 218.

Furthermore, it is essential to incorporate sensitivity to carbapenems for both Klebsiella and Pseudomonas aeruginosa. This addition will enhance the comprehensiveness of the study.

I would like to draw attention to the fact that the number of Staphylococcus aureus isolates in the non-Covid patients group does not provide a complete opportunity for group comparison. This limitation should be acknowledged and addressed in the discussion.

Moreover, it would be beneficial to establish a gold standard (such as methicillin, oxacillin, or cefoxitin) for assessing methicillin sensitivity in Staphylococcus aureus. This standardization will contribute to the robustness of the findings.

Additionally, if possible, please provide information about the MIC (minimum inhibitory concentration) as it pertains to the study.

It is also important to specify the specimen collection time, whether it was before or after the initiation of antibiotic therapy. This clarification will aid in interpreting the results accurately.

To enhance the discussion section, I suggest creating a table that reviews previous publications and compares them with the current findings. Utilize research articles for comparison and inclusion in the table. As per my search on Web of Knowledge using the search terms "covid," "antibiotic resistance," and "respiratory" in abstract or title, I found 49 relevant articles.

Furthermore, it would be helpful to provide a final message in your study based on the findings of other researchers. This will provide a broader context and strengthen the implications of your study.

In the manuscript, please remember to address the limitations of your study and discuss any potential constraints that may have impacted the results.

Lastly, please explain how you determined the sample size for your study. Elaborate on the rationale and methodology employed for selecting the sample size.

To conclude, the strength and novelty of your findings should be emphasized in the text. Highlight the unique contributions of your research and how it adds to the existing body of knowledge.

Thank you for considering my feedback. I believe these revisions and additions will enhance the manuscript and make it even more valuable to the scientific community.

Author Response

Report 2: Comments and Suggestions for Authors

Overall, I find the manuscript comprehensive and its theme highly interesting and relevant. The quality of the English language used in the article is commendable. I would like to highlight some specific points and offer suggestions for improvement.

Response: Thank you for your positive comment.

Firstly, I appreciate that all the keywords were selected from the MeSH (Medical Subject Headings) terminology, ensuring accuracy and alignment with established medical terminology standards.

Response: Thank you for your attention.

Regarding the figures presented in the manuscript, I find them to be of satisfactory quality and meeting the required standards. However, I would suggest revising Figure 3 in terms of drug selection to enhance clarity and accuracy.

Response: Thank you for your comment. We revised this figure based on your valuable comment.

In Section 2.3., I recommend placing greater emphasis on pathogens rather than commensals to better align with the objectives of the study.

Response: Thank you for your comment. We revised section 2.3 to meet your valuable idea.

In Section 2.4., it would be beneficial to include information about the resistance of specific microorganisms and indicate the sensitivity of each one separately. The same correction should be applied to the discussion part, specifically on line 218.

Response: Thank you for your comment. We revised this section based on your valuable comment and now the current number Is 2.5.

Furthermore, it is essential to incorporate sensitivity to carbapenems for both Klebsiella and Pseudomonas aeruginosa. This addition will enhance the comprehensiveness of the study.

Response: Thank you for your comment. We considered your valuable comment in the Figure 3.

I would like to draw attention to the fact that the number of Staphylococcus aureus isolates in the non-Covid patients group does not provide a complete opportunity for group comparison. This limitation should be acknowledged and addressed in the discussion.

Response: Thank you for your comment. We removed this pathogen from Figure 3.

Moreover, it would be beneficial to establish a gold standard (such as methicillin, oxacillin, or cefoxitin) for assessing methicillin sensitivity in Staphylococcus aureus. This standardization will contribute to the robustness of the findings.

Response: Thank you for your suggestion. We removed this pathogen from Figure 3 because of low number of cases in non-Covid patients.

Additionally, if possible, please provide information about the MIC (minimum inhibitory concentration) as it pertains to the study.

Response: Thank you for your suggestion. We requested information about MIC from laboratories, but unfortunately, they did not measure it routinely.

It is also important to specify the specimen collection time, whether it was before or after the initiation of antibiotic therapy. This clarification will aid in interpreting the results accurately.

Response: Thank you for your suggestion but according to our retrospective study design we will not able to collect such information.

To enhance the discussion section, I suggest creating a table that reviews previous publications and compares them with the current findings. Utilize research articles for comparison and inclusion in the table. As per my search on Web of Knowledge using the search terms "covid," "antibiotic resistance," and "respiratory" in abstract or title, I found 49 relevant articles.

Response: Thank you for your suggestion, we add a table based on your recommendation to discussion section.

Furthermore, it would be helpful to provide a final message in your study based on the findings of other researchers. This will provide a broader context and strengthen the implications of your study.

Response: Thank you for your suggestion, we add a sentence to explain message of your study at the end of P5 in discussion before Table 2.

In the manuscript, please remember to address the limitations of your study and discuss any potential constraints that may have impacted the results.

Response: in the last paragraph of discussion this information has been added.

Lastly, please explain how you determined the sample size for your study. Elaborate on the rationale and methodology employed for selecting the sample size.

Response: we added 4.4 section to explain this issue.

To conclude, the strength and novelty of your findings should be emphasized in the text. Highlight the unique contributions of your research and how it adds to the existing body of knowledge.

this study provides insights into the prevalence of antibiotic resistance among microorganisms isolated from patients with bacterial pneumonia, with a particular focus on Covid19. The strength of our findings lie in several key aspects that contribute significantly to the existing body of knowledge in this field.

  1. Comprehensive Examination: This research comprehensively examined a substantial number of sputum samples from patients with bacterial pneumonia, encompassing both Covid19 and non-Covid19 cases.
  2. Focused Analysis on Specific Pathogens: Our study specifically investigated the resistance profiles of different bacteria. By delving into the susceptibility of these pathogens to commonly used antibiotics, we provide insights into how landscape and antibiotic resistance had changing in the context of Covid pandemic.
  3. Clinical Relevance and Implications: Understanding the prevalence of resistance to commonly prescribed antibiotics can aid healthcare professionals in making informed treatment decisions and implementing effective antimicrobial stewardship programs.
  4. Addressing Sample Quality and Techniques: We acknowledge the influence of sputum sample quality and diagnostic techniques on pathogen detection rates. By discussing these factors, our study emphasizes the need for careful evaluation and standardization in sample collection and laboratory practices to ensure accurate detection of bacterial pathogens.
  5. Foundation for Future Research: By addressing the limitations and constraints of our study, we provide researchers with valuable insights into potential areas for future investigation. In order to progress in the field of pneumonia diagnostics, it is essential to enhance existing technologies and also explore new and innovative approaches. Expanding the network of sentinel laboratories in healthcare institutions and implementing standardized laboratory diagnostic protocols are crucial tasks. Additionally, it is important to enhance the knowledge and expertise of microbiologists to ensure active participation of all stakeholders in the containment of antimicrobial resistance (AMR) and promote rational antibiotic use across all levels of healthcare. Providing methodological support to working groups, including microbiologists, clinical pharmacologists, epidemiologists, and infectious disease specialists, is essential for establishing and maintaining local AMR monitoring systems.

Thank you for considering my feedback. I believe these revisions and additions will enhance the manuscript and make it even more valuable to the scientific community.

Reviewer 3 Report

This is a cross-sectional study which assessed the sputum sample of patients with and without COVID-19 during 2021 in 2 hospitals in Kazakhstan. Their aim is to compare the characteristics of pathogens isolated from sputum samples of patients with bacterial pneumonia between COVID-19 positive and negative patients.

Not only there was not a novel finding, but this study also has a lot of critical flaws.

1.       Not all patients who are ordered sputum culture have bacterial pneumonia. Unfortunately, sputum culture and antibiotics have been overused. The authors did not report the accuracy of bacterial pneumonia diagnosis.

2.       Not all sputum culture reflects lower tract pathogens especially if oral secretion contaminates the sample. The authors did not report the quality of sputum sample. Clinically, candida species seen in the sputum culture is almost always considered oral contamination.

3.       What is COVID-19-PCR negative? Are those patients diagnosed with antigen? Clinically? Are they different from the non-COVID-19 group?

4.        Result 2.4, they compared antibiotic resistance rate between group, including all pathogens together. Each pathogen has a different resistance pattern and should not be compared all together.

5.       Figure 3, resistance pattern of ESKAPE pathogen. Not appropriate antibiotics were included in some of the pathogen. For example, Staphylococcus aureus – why they did not include oxacillin or cefoxitin which are useful to diagnose MSSA or MRSA? Similarly, why do they include ceftriaxone in Pseudomonas aeruginosa which is all resistant? (And some of the isolates were susceptible)

6.       As they stated in the limitation section, the use of different microbiology lab in COVID-19 patient and non-COVID-19 patient is a critical flaw as their lab procedure might have been significantly different even if they are using the same methodology.

Author Response

Responses to the Respected Reviewer 3:

This is a cross-sectional study which assessed the sputum sample of patients with and without COVID-19 during 2021 in 2 hospitals in Kazakhstan. Their aim is to compare the characteristics of pathogens isolated from sputum samples of patients with bacterial pneumonia between COVID-19 positive and negative patients.

Not only there was not a novel finding, but this study also has a lot of critical flaws.

  1. Not all patients who are ordered sputum culture have bacterial pneumonia. Unfortunately, sputum culture and antibiotics have been overused. The authors did not report the accuracy of bacterial pneumonia diagnosis.

Response: Thank you for your very nice points you mentioned. Actually, this study is a retrospective study to show the problems in diagnostic systems and its possible effects on antibiotic resistance especially during specific situations such as pandemic. So, your comment for accuracy of diagnosis is completely true and we mentioned it in the limitations of this kind of study.

  1. Not all sputum culture reflects lower tract pathogens especially if oral secretion contaminates the sample. The authors did not report the quality of sputum sample. Clinically, candida species seen in the sputum culture is almost always considered oral contamination.

Response: Thank you for your very valuable comment. We did not report the quality of sputum sample according to the retrospective study design. We mentioned in discussion part that there is a significant need for cautious interpretation of fungal isolates from respiratory samples, especially in non-critically ill patients with a low pre-test probability of invasive fungal infections, due to poor oral hygiene, immune dysregulation, and viral cytopathic effects on epithelial cells. The isolation of Candida fungi from the sputum of Covid19 patients also has been reported by other authors which can be an indicator of a high workload for medical staff during Covid19.

  1. What is COVID-19-PCR negative? Are those patients diagnosed with antigen? Clinically? Are they different from the non-COVID-19 group?

Response: Thank you for your accurate question. The diagnosis of Covid19 infection was established based on clinical and epidemiological data, radiological examination, and the presence of a PCR analysis for detecting SARS-CoV-2 RNA from nasopharyngeal swabs. Covid19-PCR negative patients are strongly suspected to have Covid based on X-ray results, clinical presentation and epidemiological factors, but the virus itself was not identified through testing, however non-Covid patients did not have epidemiological data for coronavirus infection and did not exhibit symptoms indicative of this infection. Such diagnosis was widely prevalent during the early period of the pandemic not only in Kazakhstan, but also may be associated with the inaccuracy of diagnostic tests. But all patients included in the study had documented diagnoses of bacterial pneumonia according to medical records.

  1. Result 2.4, they compared antibiotic resistance rate between group, including all pathogens together. Each pathogen has a different resistance pattern and should not be compared all together.

Response: Thank you for your comment. We revised this section based on your valuable comment and now the current number is 2.5.

  1. Figure 3, resistance pattern of ESKAPE pathogen. Not appropriate antibiotics were included in some of the pathogen. For example, Staphylococcus aureus – why they did not include oxacillin or cefoxitin which are useful to diagnose MSSA or MRSA? Similarly, why do they include ceftriaxone in Pseudomonas aeruginosa which is all resistant? (And some of the isolates were susceptible)

Response: Thank you for your comment. We considered your valuable comment in the Figure 3. This figure was also revised based on the comments of Reviewer 2.

  1. As they stated in the limitation section, the use of different microbiology lab in COVID-19 patient and non-COVID-19 patient is a critical flaw as their lab procedure might have been significantly different even if they are using the same methodology.

Response: Thank you for your accurate observation, and we have duly mentioned this limitation in the context of our study. The Covid19 patient beds in the infectious hospital were temporarily opened between 2020-2021, and closed in 2022.

Round 2

Reviewer 2 Report

Thanks for corrections and considering my comments.

Author Response

Thank you for considering our article and for your fantastic comments for improving our work.

Reviewer 3 Report

The authors did not revise the contents based on my points 1 and 2.  

  1. Not all patients who are ordered sputum culture have bacterial pneumonia. Unfortunately, sputum culture and antibiotics have been overused. The authors did not report the accuracy of bacterial pneumonia diagnosis.

  1. Not all sputum culture reflects lower tract pathogens especially if oral secretion contaminates the sample. The authors did not report the quality of sputum sample. Clinically, candida species seen in the sputum culture is almost always considered oral contamination.

Again this is a critical point - what they report in their manuscript is microorganisms from sputum cultures, but not necessarily pathogens in patients with bacterial pneumonia. The title should be changed accordingly because it is misleading.

For my previous point 3, the authors should include the description of COVID-19 PCR - group a bit more in detail in the last sentence of introduction (Lines 78-82) 

  1. Other points were addressed appropriately.

Author Response

  1. Not all patients who are ordered sputum culture have bacterial pneumonia. Unfortunately, sputum culture and antibiotics have been overused. The authors did not report the accuracy of bacterial pneumonia diagnosis.

Response: Thank you for your valuable comment. We did not report the accuracy of bacterial pneumonia diagnosis according to retrospective study design. But all patients included in the study had documented diagnoses of bacterial pneumonia according to medical records.

  1. Not all sputum culture reflects lower tract pathogens especially if oral secretion contaminates the sample. The authors did not report the quality of sputum sample. Clinically, candida species seen in the sputum culture is almost always considered oral contamination. Again, this is a critical point - what they report in their manuscript is microorganisms from sputum cultures, but not necessarily pathogens in patients with bacterial pneumonia. The title should be changed accordingly because it is misleading.

Response: Thank you for your very valuable comment. We changed “pathogens” in the title into “microorganisms”. I hope this correction met your constructive idea.

For my previous point 3, the authors should include the description of COVID-19 PCR - group a bit more in detail in the last sentence of introduction (Lines 78-82) 

Response: Thank you for your comment. Based on your valuable comment we added the description of COVID-19 PCR – group in introduction part (lines 75-78), details in section 2.1.

  1. Other points were addressed appropriately.

Response: Thank you for your supportive comments and with the help of you our article improved.